# Analysis of miRNAs in *Osteogenesis imperfecta* Caused by Mutations in *COL1A1* and *COL1A2*: Insights into Molecular Mechanisms and Potential Therapeutic Targets

**DOI:** 10.3390/ph16101414

**Published:** 2023-10-04

**Authors:** Malwina Botor, Aleksandra Auguściak-Duma, Marta Lesiak, Łukasz Sieroń, Agata Dziedzic-Kowalska, Joanna Witecka, Marek Asman, Anna Madetko-Talowska, Mirosław Bik-Multanowski, Anna Galicka, Aleksander L. Sieroń, Katarzyna Gawron

**Affiliations:** 1Department of Molecular Biology and Genetics, Faculty of Medical Sciences in Katowice, Medical University of Silesia, 40-055 Katowice, Poland; aaugusciak@sum.edu.pl (A.A.-D.); mlesiak@sum.edu.pl (M.L.); lukasz.sieron@sum.edu.pl (Ł.S.); alsieron@sum.edu.pl (A.L.S.); 2Department of Parasitology, Faculty of Pharmaceutical Sciences in Sosnowiec, Medical University of Silesia, 41-200 Sosnowiec, Poland; jwitecka@sum.edu.pl; 3Department of Medical and Molecular Biology, Faculty of Medical Sciences in Zabrze, Medical University of Silesia in Katowice, 41-808 Zabrze, Poland; masman@sum.edu.pl; 4Department of Medical Genetics, Jagiellonian University Medical College, 30-663 Krakow, Poland; a.madetko-talowska@uj.edu.pl (A.M.-T.); miroslaw.bik-multanowski@uj.edu.pl (M.B.-M.); 5Department of Medical Chemistry, Medical University of Bialystok, 15-222 Bialystok, Poland; anna.galicka@umb.edu.pl

**Keywords:** *Osteogenesis imperfecta*, skin fibroblasts, extracellular matrix, collagen, mutation, miRNAs

## Abstract

*Osteogenesis imperfecta* (*OI*) is a group of connective tissue disorders leading to abnormal bone formation, mainly due to mutations in genes encoding collagen type I (Col I). Osteogenesis is regulated by a number of molecules, including microRNAs (miRNAs), indicating their potential as targets for *OI* therapy. The goal of this study was to identify and analyze the expression profiles of miRNAs involved in bone extracellular matrix (ECM) regulation in patients diagnosed with *OI* type I caused by mutations in *COL1A1* or *COL1A2*. Primary skin fibroblast cultures were used for DNA purification and sequence analysis, followed by analysis of miRNA expression. Sequencing analysis revealed mutations of the *COL1A1* or *COL1A2* genes in all *OI* patients, including four previously unreported. Amongst the 40 miRNAs analyzed, 9 were identified exclusively in *OI* cells and 26 in both *OI* patients and the controls. In the latter case, the expression of six miRNAs (hsa-miR-10b-5p, hsa-miR-19a-3p, hsa-miR-19b-3p, has-miR-204-5p, has-miR-216a-5p, and hsa-miR-449a) increased, while four (hsa-miR-129-5p, hsa-miR-199b-5p, hsa-miR-664a-5p, and hsa-miR-30a-5p) decreased significantly in *OI* cells in comparison to their expression in the control cells. The identified mutations and miRNA expression profiles shed light on the intricate processes governing bone formation and ECM regulation, paving the way for further research and potential therapeutic advancements in *OI* and other genetic diseases related to bone abnormality management.

## 1. Introduction

*Osteogenesis imperfecta (OI*) is a group of genetically related connective tissue disorders, diverse in terms of both genotypes and phenotypes, characterized by increased bone fragility. Clinically, five types of *OI* ranging from mild (type I) to moderate (types III and IV) and severe or lethal (type II) are distinguished [1,2,3]. So far, numerous mutations in genes encoding collagen type I (Col I) or involved in collagen modification, e.g., cartilage-associated protein (CRTAP) and leucine proline-enriched proteoglycan 1 (LEPRE1) [4,5]; folding, e.g., serpin peptidase inhibitor (SERPINH1) [6]; and processing, e.g., bone morphogenetic protein 1 (BMP1) [7], have been shown to cause various types of *OI*. Nevertheless, a majority of *OI* cases inherited in an autosomal-dominant manner are caused by mutations in the collagen type I α1 or α2 encoding genes (*COL1A1* and *COL1A2*, respectively), resulting in a reduced amount or abnormal structure of collagen [8,9,10].

The earliest stage when an *OI* diagnosis can be made is the prenatal period. It depends on the severity of the *OI* and is based on family history; genetic tests, i.e., sequencing of genomic DNA and specific mutation identification; and prenatal tests, i.e., a decrease in the femoral length (FL) [11]. In the post-natal period, positive family history and pivotal clinical symptom (bone fragility, typical features of blue sclera) notification with bone density measurement (BMD) implementation are routinely used in *OI* diagnosis [12,13]. Additional diagnostic methods include magnetic resonance imaging (MRI), radiography of the uterus, and other genetic tests, such as multiplex ligation-dependent probe amplification (MLPA) or chorionic villus sampling (CVS) [12,14]. If the typical clinical symptoms and positive family history are noted by the clinician, routine diagnostic methods are sufficient to diagnose *OI*; however, about 10–15% of patients suffer from less common variants of the disease. Moreover, any diagnostic method used may be clear-cut and have drawbacks, e.g., employment of the FL carries the risk of other skeletal dysplasia’s recognition instead of *OI* [13,15]. In such circumstances, further procedures, mostly based on molecular diagnostics, are needed [11].

MicroRNA (miRNA) constitutes short, about 20-nucleotide, single-stranded, and endogenous RNA molecules that have been observed to negatively regulate gene expression in two manners: i) by attaching to their target messenger RNA (mRNA) at different mRNA sites, most often 100% complementary, and cutting it; or ii) by binding to the 3′-untranslated region (3′UTR) of the target mRNA and inhibiting translation, an action that does not require 100% complementarity [16]. The principal role of miRNAs is the regulation of a number of biological processes, such as embryogenesis, apoptosis, hosting immune responses during infections, or cancerogenesis [16]. miRNAs fulfill a crucial role in the regulation of gene expression and are involved in bone formation and bone diseases [17,18,19]. For instance, miR-217 has been reported to promote cell proliferation and osteogenic differentiation during the development of steroid-associated osteonecrosis in bone marrow mesenchymal stem cells [20,21], whereas miR-200a-3p has been linked with the synthesis of ECM components in fibrotic diseases [22].

Over the past two decades, the mechanisms of human extracellular matrix (ECM) regulation by miRNAs have been extensively studied, including the genes involved in collagen synthesis. For example, the negative influence of hsa-miR-133a-3p, which has been assigned a possible antifibrotic role [23,24]; hsa-miR-27b-3p, which increases collagen expression [25]; and hsa-miR-382-5p, which promotes epidural fibrosis [26,27], has been demonstrated. Furthermore, many miRNAs that can regulate genes involved in the enzymatic processing or degradation of natively synthesized collagen have been recently discovered. For example, hsa-miR-377-3p has been shown to inhibit the synthesis of matrix metalloproteinase-2 (MMP-2) and MMP-16, whereas hsa-miR-200b-3p has been proven to regulate tissue inhibition of metalloproteinase-2 (TIMP2) [26]. Moreover, miRNAs can affect signaling pathways involved in ECM synthesis and degradation; e.g., hsa-miR-216a-5p has been shown to regulate Janus kinase 2 (JAK2) [28,29], while hsa-miR-25-3p has been shown to control the transforming growth factor β receptor I (TGFβR1) and neurogenic locus notch homolog protein 1 (NOTCH1) signaling pathways [30,31]. To date, however, only a few miRNAs, such as let-7a-5p, miR-92a, and miR-29b, have been studied in *OI* [17,18].

The aim of this study was to identify and analyze the expression profiles of miRNAs in osteogenesis imperfecta (*OI*) type I caused by mutations in the *COL1A1* and *COL1A2* genes.

## 2. Results

### 2.1. Mutation Analysis in the COL1A1 and COL1A2 Genes from OI Type I

*Osteogenesis imperfecta* type I in the majority of cases (~95%) is caused by mutations in the *COL1A1* and *COL1A2* genes; hence, in this study, we focused on sequencing analyses of those genes [1,2]. Mutations in *COL1A1* were found in 8 study subjects and in *COL1A2* in 2 out of 10 study subjects diagnosed with *OI* type I. The missense mutations were detected in both *COL1A1* (donor nos. 1–4) and *COL1A2* (donor nos. 9 and 10). In *COL1A1*, other mutation types were also found, including a single-nucleotide deletion resulting in a frameshift in three donors (nos. 5–7) and an exon skipping in one patient (no. 8). The missense mutations detected in the *COL1A1* gene resulted in three different substitutions, namely glycine by alanine in exon 17 (patient no. 1), glycine by valine in exon 9 (patient no. 2), and a third one also found in exon 17 that resulted in valine-to-phenylalanine alteration (patient no. 3). Additionally, in two other patients (nos. 6 and 7), two deletions of T located in exon 30 and exon 17 of the *COL1A1* gene resulted in the production of an altered protein. In contrast, both missense mutations in the *COL1A2* gene resulted in the substitution of glycine by serine in exon 31 (patient no. 9) or exon 38 (patient no. 10).

Moreover, we identified a series of new mutations in both genes: substitution of guanine for thymine in exon 9 in patient no. 2 and deletion of 10 nucleotides in exon 50 (frameshift) adjacent to a missense mutation (substitution of glutamine by valine) in *OI* patient no. 4, both in the *COL1A1* gene. Another alteration comprised a guanine deletion in exon 2 of the *COL1A1* gene and was identified in patient no. 5. A new mutation found in the *COL1A2* gene was represented by a substitution of guanine for adenine in exon 31 of patient no. 9.

A brief clinical description of *OI* patients harboring previously reported mutations as well as the new mutations in *COL1A1* and *COL1A2* identified in this study are presented in Table 1.

### 2.2. Identification of miRNAs in OI Type I

First, we performed identification analyses of the miRNAs in the *OI*-derived cells and control cells. As shown in Table 2, the results were categorized into the three following parts: (A) miRNAs detected in *OI* but not in the control fibroblasts; (B) miRNAs detected in both *OI* and the control cells; (C) miRNAs not detected in *OI* nor the control cells.

As presented in Table 2, nine specific miRNAs, i.e., hsa-miR-19a-3p, hsa-miR-19b-3p, hsa-miR-133a-3p, hsa-miR-200b-3p, hsa-miR-204-5p, 216a-5p, hsa-miR-377-3p, hsa-miR-449a, and hsa-miR-590-5p, were identified in the fibroblasts of 30% to 70% of *OI* donors, but not in the control fibroblasts. Other 26 miRNA molecules were detected in both the *OI* and the control cells, with the percentage of *OI* donors with a specific miRNA detection ranging from 70% to 100%. Finally, five miRNAs selected for analysis, i.e., hsa-miR-1-3p, hsa-miR-141-3p, hsa-miR-142-3p, hsa-miR-200a-3p, and hsa-miR-217, were not detected in the *OI* or the control cells.

### 2.3. Expression Analysis of miRNAs in OI

Based on the miRNA identification, an expression analysis was performed for miRNAs detected in >50% of *OI* patients and compared with the controls. As depicted in Figure 1, increased expression was observed in the case of six miRNAs (hsa-miR-10b-5p, hsa-miR-19a-3p, hsa-miR-19b-3p, has-miR-204-5p, has-miR-216a-5p, and hsa-miR-449a), while decreased expression was observed in the case of four miRNAs (hsa-miR-129-5p, hsa-miR-199b-5p, hsa-miR-664a-5p, and hsa-miR-30a-5p) in *OI* as compared to the controls. Among the miRNAs analyzed, miR-449a’s expression increased about 20-fold, followed by an almost 5-fold increase in miR-19b-3p and a ~4-fold increase in miR-19a-3p. Meanwhile, an almost 6-fold decrease in expression in *OI* was found in the case of miR-664a-5p, whereas a slightly less decrease was observed for miR-129-5p (3.6-fold), miR-199b-5p (2.8-fold), miR-30a-5p (2.2-fold), and miR-25-3p (1.9-fold) (Table 3).

### 2.4. miRNA Correlation Analysis in OI

Among the miRNAs that were overexpressed in the *OI* cells, two sets displaying moderate-to-strong positive correlations within each group were observed (Table 4). The first group consisted of miR-10a-5p, miR-10b-5p, miR-199a-5p, and miR-382-5p, and the second consisted of miR-19a-3p, miR-19b-3p, miR-204-5p, miR-216a-5p, and miR-449a. miR-17-5p, on the other hand, did not show strong correlations with any of the overexpressed miRNAs.

A significantly decreased expression in the *OI* group was noted for 13 miRNAs (Table 5). In this group, an almost complete correlation was found for miR-21-5p, miR-25-3p, miR-26a-5p, miR-26b-5p, miR-27a-3p, miR-27b-3p, miR-29a-3p, and miR-29c-3p, which have been previously reported to regulate the expression of genes associated with fibrosis (cystic fibrosis in children or liver fibrosis) and cancerogenesis [24,25,48,49,56]. Further, miR-30a-5p showed a strong correlation with almost all miRNAs presented in Table 5.

In contrast, a moderately negative correlation of miR-129-5p with miR-21-5p, miR-26b-5p, miR-29a-3p, and miR-29c-3p was observed. This may indicate an antagonistic function of miR-129-5p in the regulation of collagen synthesis. Finally, the weakest correlations with other tested downregulated miRNAs were observed for miR-199b-5p and miR-29b-3p (Table 5).

### 2.5. Association of Identified miRNAs with Mutations in COL1A1 and COL1A2

In the group of *OI* patients harboring mutations in *COL1A1*, upregulated expression of miR-17-5p, miR-21-5p, miR-26b-3p, miR27a-3p, miR27b-3p, miR29a-3p, mir29c-3p, miR-30a-5p, miR-34a-5p, miR-92a-3p, and miR143-3p was found. In contrast, in the *OI* caused by mutations in *COL1A2*, only miR-25-3p was found to be upregulated (Table 6).

Interestingly, the expression of miR-132-3p was comparable in the controls and the *COL1A1*-mutated fibroblasts but increased in the group of *OI* caused by mutations in *COL1A2* (Table 6). These observations may suggest that miR-25-3p and miR-132-3p could play a particularly significant role in the pathogenetic processes of *OI* induced by mutations in the *COL1A2* gene.

Considering the type of mutation, it was noted that patients no. 1, 2, and 3 with missense mutations in the *COL1A1* gene revealed similar overexpression patterns for several miRNAs, indicating that these mutations may have similar effects on miRNA regulation. Patients no. 4, 6, and 7, with frameshift mutations in the *COL1A1* gene, displayed similar overexpression patterns for some miRNAs but also individually dependent differences among particular types of miRNAs, and the profiles of expression also varied from those found in *OI* with missense mutations. In turn, patient no. 5, with a frameshift mutation in the *COL1A1* gene, had a unique expression pattern of overexpressed and downregulated miRNAs. In summary, the miRNA expression patterns observed in the individually analyzed *OI* patients indicate that particular mutations may influence the regulation of specific miRNAs, potentially leading to downstream effects on genes involved in bone development and maintenance (Table 6).

## 3. Discussion

Despite similar and uniform phenotypic symptoms, the molecular mechanisms of pathogenetic processes contributing to *OI* remain unclear. This disorder is difficult to treat as it can be caused not only by direct defects in the amount of type I collagen produced but also by the influence of various proteins and regulatory RNA molecules, like miRNAs, on the processing of synthesized collagen, including the pool of aberrant proteins.

Around 5% of *OI* cases have an unknown genetic background. DNA sequencing analyses revealed the presence of mutations in both genes of all *OI* donors enrolled in this study, including four novel and previously unreported, of which three were identified in the *COL1A1* gene and one in the *COL1A2* gene. The missense mutations detected in the *COL1A1* and *COL1A2* genes resulted in the substitution of glycine and other amino acids, leading to disruptions in the collagen chain interactions and destabilization of the collagen triple helix [1,76]. Frameshift mutations, including single-nucleotide deletion, ten-nucleotide deletion, and exon skipping in the *COL1A1* gene, altered the reading frame, resulting in truncated or absent Col I chains. Additionally, a single-nucleotide deletion in exon 2 of *COL1A1* disrupted the formation of the collagen triple helix. These mutations likely contribute to the development of *OI* by affecting the stability and packing of the collagen molecule [77]. The mutations identified in this study align with previous reports on *OI*, reaffirming that mutations in the *COL1A1* and *COL1A2* genes are responsible for the majority of *OI* cases [78,79].

This research focused on the investigation of the miRNAs that regulate the expression of the ECM, including Col I processing and trafficking in the context of *OI* pathogenesis. Nine miRNAs that were identified in *OI* but not in the controls, namely hsa-miR-19a-3p, hsa-miR-19b-3p, hsa-miR-133a-3p, hsa-miR-200b-3p, hsa-miR-204-5p, 216a-5p, hsa-miR-377-3p, hsa-miR-449a, and hsa-miR-590-5p, have been previously reported to regulate the TGF-β and Wnt signaling pathways, indicating their essential role in the regulation of genes involved in bone formation and turnover [28,29,32,33,34].

Five out of forty analyzed miRNAs, i.e., hsa-miR-1-3p, hsa-miR-141-3p, hsa-miR-142-3p, hsa-miR-200a-3p, and hsa-miR-217, were absent in the *OI* group and the controls. These molecules affected suppressor genes, such as the phosphatase and tensin homolog (*PTEN)* and the dickkopf WNT signaling pathway inhibitor 1 (*DKK1)*, as well as genes involved in ECM turnover, i.e., Rho-associated coiled-coil-containing protein kinase 2 (*ROCK2)* [20,21,22,71,72,73,74,75]. These miRNAs have not been directly linked to genes related to collagen synthesis and secretion or connective tissue diseases; however, they participate in the inhibition of the epithelial–mesenchymal transition in renal tubular epithelial cells (e.g., miR-141-3p) and the promotion of cell proliferation and osteogenic differentiation during the development of steroid-associated osteonecrosis in bone marrow mesenchymal stem cells (e.g., miR-217) [21,73] (Table 2). Moreover, hsa-miR-200a-3p has been found to participate in the overproduction of ECM proteins in fibrotic diseases.

The involvement of miRNAs in ECM regulation has recently become an area of intense research, as their ability to modulate gene expression post-transcriptionally makes them potent regulators of crucial cellular processes [17,18,19]. Understanding how these miRNAs affect ECM components will be of great importance in the search for new therapeutic targets for therapies for diseases associated with ECM dysregulation.

The expression analysis performed herein showed altered expression of 10 miRNAs in the fibroblasts of *OI* patients compared to the controls. The expression of six, i.e., hsa-miR-10b-5p, hsa-miR-19a-3p, hsa-miR-19b-3p, has-miR-204-5p, has-miR-216a-5p, and hsa-miR-449a, was substantially increased, while the expression of four (hsa-miR-129-5p, hsa-miR-199b-5p, hsa-miR-664a-5p, and hsa-miR-30a-5p) decreased.

Further, the fold change analysis provided robust validation of the observed miRNA expression alterations in *OI* compared to the controls, displaying a significant multiplication in up- or downregulation. Remarkably, miR-449a’s expression increased about 20-fold, while miR-449a was exclusively detected in *OI* patients, which may suggest the potential relevance of both miRNAs in the pathogenesis of *OI*.

In contrast, we noted a six-fold expression decrease in miR-664a-5p. Though this miRNA has been implicated in the modulation of tumor suppressor p53 and the expression of p53 target genes and has recently been associated with osteosarcoma [70], this finding adds an intriguing dimension to this study, hinting at potential links between *OI* and p53-related pathways that warrant further investigation.

The intricate relationships between miRNAs may offer a unique opportunity to establish connections between molecular processes, providing insights into the pathogenesis of diverse diseases. For better clarification of the impact of miRNAs on bone ECM changes in *OI*, we explored their potential correlations, whether positive or negative. Correlation analyses were carried out within the groups with upregulated and downregulated expressions of miRNAs. The miRNAs overexpressed in *OI* that strongly correlate with each other are involved in the regulation of osteoblast differentiation and osteogenesis [26,27,44,45,46,66,67], while the miRNAs presented in the latter group have been previously associated with various biological processes and diseases, including cancer, inflammation, and neuronal differentiation [28,29,32,33,34,36,37,38]. With the exception of miR-129-5p, which showed a moderate negative correlation with four other miRNAs associated with various cancers, both groups of analyzed miRNAs revealed positive correlations with miR-29 family members [49,52,56]. The miR-29 family negatively regulates collagen expression by directly targeting *COL1A1*’s and *COL1A2*’s mRNA transcripts, while the other miRNAs are linked to fibrosis and cancers, suggesting their effect on the synthesis of the ECM [30,31,48,49,50,51].

In the present study, 40 miRNAs in total were analyzed in 10 *OI* donors in whom mutations, including four unreported ones, were identified in *COL1A1*/*COL1A2*. Overexpression of miR-10b-5p and miR-382-5p in some patients suggests their involvement in the pathogenesis of *OI*. On the other hand, decreased expression of miR-129-5p, miR-199b-5p, miR-29b-3p, and miR-664a-5p in most patients suggests their importance in normal bone development and function (Table 6). Kaneto et al. reported decreased expression of miR-29b during the osteoblastic differentiation of mesenchymal stem cells (MSCs). Likewise, in our research, we observed a modest decrease in the expression of miR-29b in skin fibroblasts from a group of patients diagnosed with *OI* type I. Upon conducting separate analyses for each *OI* patient, we observed an upregulation of miR-29b expression in certain individuals, which contrasts with the overall trend observed in the entire *OI* group. This variability may be attributed to the diverse range of patient demographics, including age and gender. miR-29 family members, including miR-29a, miR-29b, and miR-29c, have been implicated in the regulation of ECM proteins, such as collagen, which are important for bone development and homeostasis [52,53,54,55,56]. The aberrant regulation of those miRNAs in patients with *OI* type I induced by mutations in *COL1A1* may contribute to reduced collagen synthesis and impaired bone formation. Regarding *OI* patients harboring mutations in the *COL1A2* gene, we observed overexpression of miRNA-25-3p. Noticeably, this miRNA has been implicated in the regulation of bone mineralization and may play a role in osteoblast differentiation [30,31]. We also observed some discrepancies in the expression of certain miRNAs, such as let-7a, which was found to be overexpressed in some patients with missense mutations, but its expression was downregulated in patients with an exon-skipping mutation. Interestingly, one patient (no. 5) with a novel identified mutation that resulted in a frameshift in exon 2 of the *COL1A1* gene displayed a distinct miRNA expression pattern compared to other examined patients, which may suggest a unique background of *OI* in this individual.

One of the potential therapies for *OI* based on targeting the miRNAs involved in the regulation of bone formation and remodeling can be applied by either increasing the expression of beneficial miRNAs or decreasing the expression of harmful ones [80,81,82]. However, there are several challenges that need to be addressed before its implementation into clinical practice, including: (i) the identification of disease-specific and most effective targeted miRNAs (as discussed in the current work); (ii) the validation of relevant delivery methods that would not only effectively “reach” the targeted micromolecule but also efficiently deliver miRNAs to the bone; and (iii) the overcoming of the immune response to exogenous miRNAs [83].

## 4. Conclusions

Despite similar and uniform phenotypic symptoms in individuals with *OI*, the molecular basis of this disease seems to be varied and still unclear. Moreover, it is difficult to treat, as it can be caused not only by direct defects in the amount of type I collagen produced but also by the influence of various proteins and regulatory RNA molecules, including miRNAs, on the processing of synthesized collagen and the pool of aberrant proteins. Therefore, understanding the function of miRNAs that can affect collagen or signaling pathways that indirectly regulate the amount or quality of collagen is extremely important and necessary. Overall, the current study underscores the importance of the ECM in the development of *OI* and other genetic diseases of connective tissue. The identified mutations and miRNA expression profiles shed light on the intricate processes governing bone formation and ECM regulation, paving the way for further research and potential therapeutic advancements in *OI* and other genetic diseases related to bone abnormality management.

## 5. Materials and Methods

### 5.1. Study Participants and Control Material

This study was carried out in accordance with the Declaration of Helsinki and was approved by the Bioethics Committee of the Jagiellonian University in Krakow, Poland (protocol no. DK/KB/CM/0031/689/2010). Clinical examination and medical histories of the subjects were taken by a clinician from the Chair of Pediatrics, Department of Medical Genetics, UJCM, Krakow, Poland. All subjects (or representing individuals) read and signed a written informed consent form prior to inclusion in this study.

Ten subjects (ranging in age from 0.5 to 29 years, of both sexes, and diagnosed with *OI* type I) were enrolled in this study. The diagnosis was confirmed on the basis of multiple (from ~9 to 20) bone fractures in the post-natal period and/or blue sclera or osteoporosis. The inclusion criterion was a clinically confirmed diagnosis of *OI* type I. Exclusion criteria included the clinical diagnosis of other types of *OI* and absence of mutations in the *COL1A1* and/or *COL1A2* genes. A clinical and molecular description of the *OI* type I subjects enrolled in this study is presented in Table 1. As a control, the human fibroblast cell line BJ (CRL-2522, ATCC, Manassas, VA, USA) was used.

### 5.2. Isolation Procedure of Skin Fibroblasts from OI Donors

Skin biopsy samples were collected from 10 patients diagnosed with *OI* type I. To establish a fibroblast culture, skin biopsies of a size of about 3–5 mm^2^ were cut into smaller pieces and placed in a Petri dish with 10 mL of Dulbecco’s Modified Eagle’s Medium (DMEM), supplemented with 4.5 g/L glucose, 10% fetal calf serum (FCS), 50 μg/mL streptomycin, 50 U/mL penicillin, and 25 µg/mL amphotericin B (PAA, Austria). After one week, the remaining parts of the skin explants were removed from the dish, and the fibroblast culture was kept until 90% confluence. Subsequently, the cells were harvested using trypsin/EDTA, plated in fresh DMEM with 4.5 g/L glucose, 10% FCS, streptomycin, penicillin, and amphotericin B in plastic flasks, and kept until homogeneous cultures were obtained. The homogeneity of the fibroblast culture was confirmed after first passage by immunofluorescent staining against vimentin. Cell viability amounted to 98% using Trypan blue.

### 5.3. Fibroblast Cultures

For the experiments, 2 × 105 cells per well were plated in 6-well culture plates and maintained for 7 days in DMEM with 4% FBS supplemented every second day with a fresh L-ascorbic acid 2-phosphate sesquimagnesium salt hydrate (Sigma-Aldrich, St. Louis, MO, USA) at a concentration of 40 μg/mL. The experiments were carried out between the third and sixth passages. Control human fibroblast cell line BJ (CRL-2522, ATCC, Manassas, VA, USA) was cultured and treated identically as the primary skin fibroblasts recovered from the study group.

### 5.4. Fibroblast DNA Isolation and Amplification

Lysis and DNA isolation from primary skin fibroblasts and the control cell line were conducted according to the manufacturer’s protocol using DNA purification kit Blood Mini (A&A Biotechnology, Gdansk, Poland). The DNA amplification reactions were conducted using FastStartTaq DNA polymerase kit (Roche, Basel, Switzerland). The composition of the reaction mixture and conditions used are described elsewhere [84,85]. Briefly, for each gene, 10 amplification reactions were conducted, and the DNA fragments were visualized in 1% agarose gel (Agarose, LE Analytical Grade, Promega, Madison, WI, USA) in TAE buffer.

### 5.5. Sequence Analyses of COL1A1 and COL1A2 Genes from OI Donors

Sequencing PCR for the modified Sanger’s enzymatic method was conducted according to the recommendation provided by the manufacturer of the ABI Prism 3130xl DNA Analyzer. In brief, 20 ng of each purified amplified PCR product was mixed with 5 pM of the appropriate sequencing primer and Big Dye Terminator v.3.1, as previously described [84,85]. Before sequencing analysis, DNA fragments were purified employing the BigDye XTerminator Purification Kit (Applied Biosystems, Waltham, MA, USA) according to the manufacturer’s manual. Bioinformatic analysis of the DNA sequences obtained was conducted by the Chromas Lite 2.01 and NCBI Nucleotide Blast software against the *COL1A1* (NG_007400.1) and *COL1A2* (NG_007405.1) genes as the reference sequences.

### 5.6. Fibroblast RNA Isolation

Primary skin fibroblasts and control cell line cultures were harvested with trypsin/EDTA (PAA Laboratories, Colbe, Germany) and washed twice with sterile phosphate buffered saline (PBS) (PAA Laboratories, Colbe, Germany). The cell amount was counted using Trypan Blue (BIO-RAD, Hercules, CA, USA) in the TC-20 Automated Cell Counter (BIO-RAD, Hercules, CA, USA). Cell lysis and purification of total RNA and miRNA were conducted using 2.5 × 106 cells in total. The isolation procedure was performed with the miRNeasy Mini Kit (Qiagen, Hilden, Germany) according to the manufacturer’s instructions. The purity and concentration of the collected RNA samples were determined using a Nanodrop 2000 spectrophotometer (Thermo Fisher Scientific, Waltham, MA, USA).

### 5.7. Real-Time Quantitative Polymerase Chain Reaction

The miScript II RT Kit (Qiagen, Hilden, Germany) and an RNA concentration of 250 ng were used to synthesize cDNA. The obtained cDNA was used to examine the defined miRNA expression (Table 2). For this purpose, expression analysis of Custom miScript miRNA PCR Array, conf. no. CMIHS02722F (Qiagen, Hilden, Germany), was performed in a thermal cycler, the LightCycler 480 II (Roche, Basel, Switzerland). The quantitative polymerase chain reaction analysis was carried out in quadruplicate with miScript SYBR Green PCR Kit (Qiagen, Hilden, Germany) according to the manufacturer’s manual.

### 5.8. Statistical Analyses

The expression analysis of miRNAs was performed by GenEx ver6 software (MultiD Analyses AB). Raw data were normalized to the reference genes (*SNORD61*, *SNORD68*, *SNORD72*, *SNORD95*, *SNORD96A*, and *RNU6-2*) according to the manufacturer’s protocol. The Kolmogorov–Smirnov test was employed to determine the distribution of the data (Appendix A). For miRNA data that presented normal distribution, a parametric, unpaired, one-tailed t-test was conducted. In the case of miRNAs that showed non-normal distribution of data, a one-tailed Mann–Whitney test was used. The results were considered statistically significant at *p* < 0.05. The correlation analyses of miRNAs were performed using the Spearman method. Comparative expression of miRNAs in the study group and the control cell line was determined by scatter plot analysis.

## Figures and Tables

**Figure 1 pharmaceuticals-16-01414-f001:**
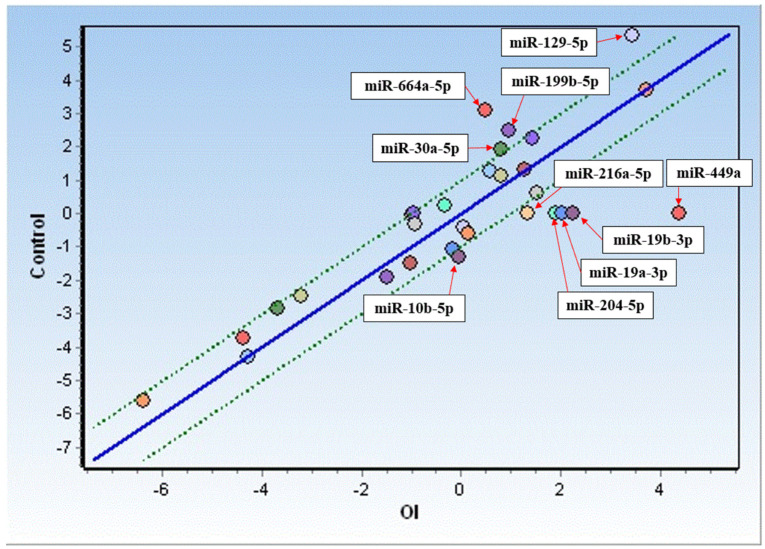
Comparison of miRNA expression in fibroblasts from *OI* donors (*OI*) and control cells. Only upregulated (below the green line) and downregulated (above the green line) miRNAs have been presented.

**Table 1 pharmaceuticals-16-01414-t001:** Brief clinical summary and characterization of mutations detected in the *COL1A1* and *COL1A2* genes of *OI* donors enrolled in this study.

Patient No.	Age (Yrs)	Sex (F/M)	RefSNP (rs) Number or Mutation Position	Type of Mutation	Brief Clinical Description
*OI* donors harboring mutations in the *COL1A1* gene					
1	10	M	rs1907512918	Missense	About 10 bone fractures, osteoporosis (densitometry).
2	0.5	M	g.8874 G/T; c.662 G/T; p.Gly221Val; exon 9 ***	Missense	Right femur fracture in early infancy, motor development delay, joint hypermobility, funnel chest, submucosal cleft palate, blue sclera, negative family history.
3	8	F	rs72645362	Missense	About 11 bone fractures, normal hearing, positive family history.
4	4	F	g.20199 A/T, g.202000 delGACTGGTGAG; c.3881 A/T, c.3882-3891 del; p.Glu1294ValfsX32; exon 50 ***	Missense and 10-nucleotide deletion (frameshift)	About 9 bone fractures, normal hearing, speech defect, normal bone density, lack of bone deformities, blue sclera, negative family history, no dentinogenesis.
5	20	F	g.2113 delG, frameshift; c.231 delG; p.Thr78Profs; exon 2 ***	Single-nucleotide deletion (frameshift)	Blue sclera, negative family history.
6	10	F	rs72651634	Single-nucleotide deletion (frameshift)	About 12 bone fractures, normal hearing.
7	5	M	rs72645370	Single-nucleotide deletion (frameshift)	About 2 bone fractures per year (at the observation period), blue sclera, negative family history.
8	29	M	rs72653156	Exon skipping	About 20 bone fractures, normal hearing, osteoporosis (densitometry).
*OI* donors with mutations in the *COL1A2* gene					
9	13	M	g.26908 G/A; c.1828 G/A; p.Gly610Ser; exon 31 ***	Missense	N.D.
10	15	M	rs72658185	Missense	N.D.

* New mutations reported in the COL1A1/COL1A2 genes from *OI* type I; F—female; M—male; N.D.—no data available.

**Table 2 pharmaceuticals-16-01414-t002:** Characteristics of analyzed miRNAs.

miRNAs	Percentage of *OI* Donors with a Specific miRNA Detection	Target Gene(s)	Effect(s)	Location and/or Characteristics of miRNAsin ECM-Related Diseases	Ref.
PART A	miRNAs detected in *OI* but not in control fibroblasts				
hsa-miR-19a-3p	70%	*TGFβRII*	Not found	- Inhibition of HCF autophagy by targeting TGF-β R II mRNA; - Human cardiac fibroblasts (HCFs).	[32]
hsa-miR-19b-3p	70%	*TGFβRII*	Not found	- Inhibition of HCF autophagy by targeting TGF-β R II mRNA; - Prostate cancer; - Human cardiac fibroblasts (HCFs).	[33,34]
hsa-miR-133a-3p	30%	*COL1A1*	Negative	- Myocardial and liver fibrosis; - Possible antifibrotic role; - Hepatic stellate cells.	[23,24]
hsa-miR-200b-3p	30%	*TIMP2*	Negative	- Endometrial cancer; - Associated with lncRNAMEG3/hsa-miR-200b-3p/AKT2 pathway in osteosarcoma.	[26,35]
hsa-miR-204-5p	50%	*MMP9*	Not found	- Inhibition of cell proliferation, migration, and invasion; - Promotion of cell apoptosis in malignant melanoma.	[36]
hsa-miR-216a-5p	50%	*JAK2*	Negative	- Osteoarthritis and endometriosis; - Association with changes in the estrogen/H19/MIR-216A-5P/ACTA2 pathway in endometriosis; - Proliferation, migration, and inhibition of chondrocyte apoptosis by MIRA-5P/JAK/STAT3 signaling pathway; - Chondrocyte.	[28,29]
hsa-miR-377-3p	30%	*MMP2* and *MMP16*	Negative	- Endometrial cancer; - Esophageal cancer.	[26]
hsa-miR-449a	70%	*LEF1*	Not found	- Chondrogenesis of human mesenchymal stem cells (hBM-MSCs); - Human bronchial epithelial cells.	[37,38]
hsa-miR-590-5p	30%	*FGF18*	Negative	- Affection of chondrocyte proliferation, apoptosis, and inflammation in osteoarthritis; - Increased expression with atrial fibrillation; - Possible contribution in collagen production and fibrosis.	[39,40]
PART B	miRNAs detected in *OI* and control cells				
hsa-let-7a-5p	100%	Not found	Not found	- Liver-specific overexpressed miRNA in plasma of human immunodeficiency virus (HIV)/hepatitis C virus (HCV) patients.	[41]
hsa-let-7d-5p	100%	*HMGA1*	Negative	- Decreased expression in ovarian cancer; - Inhibition of proliferation, apoptosis induction, and chemosensitivity of ovarian cancer cells to cisplatin by silencing *HMGA1* (via the p53 signaling pathway).	[42,43]
hsa-miR-10a-5p	90%	Not found	Not found	- Tissue-specific expression, e.g., heart, ovaries, and testes; - Decreased expression in small intestine, lung, and spleen; - Increased expression in glioblastoma, anaplastic astrocytomas, primary hepatocellular carcinomas, and colon cancer; - Regulation of type I collagen in hypertrophic scars.	[44]
hsa-miR-10b-5p	90%	*MDM2* and *MDM4*	Negative	- Tissue-specific expression, e.g., muscles and liver; - Decreased expression in skeletal muscles; - Increased expression detected in glioblastoma, anaplastic astrocytomas, pancreatic cancer, and metastatic breast cancer; - Stomach cancer; - Epithelial–mesenchymal transition in breast cancer.	[45,46]
hsa-miR-17-5p	70%	*circMTO1, Smad7*	Negative	- Pro-fibrotic inductor in liver fibrosis; - Oncogenic miRNA (cancer development).	[47]
hsa-miR-21-5p	100%	SPRY1/ERK/NF-kB pathway	Not found	- Liver and lung fibrosis progressor (SPRY1/ERK/NF-kB signaling pathway activation); - Obstructive renal fibrosis.	[48]
hsa-miR-25-3p	90%	*TGFBR1, ADAM17, FKBP14,* NOTCH1 pathway	Negative	- Cystic fibrosis in children.	[30,31]
hsa-miR-26a-5p	100%	*LOXL2*	Negative	- Head and neck squamous-cell carcinoma.	[49]
hsa-miR-26b-5p	100%	*LOXL2*	Negative	- Renal cell carcinoma; - Head and neck squamous-cell carcinoma.	[49]
hsa-miR-27a-3p	100%	*TMBIM6*	Negative	- Diabetic nephropathy; - Reduction in renal fibrosis (TGF-β, fibronectin, collagen I and II, and actin III); - Osteoporosis (probable therapeutic target and diagnostic biomarker).	[50,51]
hsa-miR-27b-3p	90%	*COL1A1*, Adamts8, *THBS1*, *FN1*, *COL5A1*, and *COL14A1*	Positive	- Osteoarthritis.	[25]
hsa-miR-29a-3p	100%	Not found	Not found	- Inhibition of fibroblast-derived collagen I production; - Stomach cancer; - COL3A1/FBN1/COL5A2/SPARC-MIR-29A-3P-H19.	[52,53]
hsa-miR-29b-3p	80%	*STAT3* Track H19/miR-29b-30/SOX9 pathway	Negative on *STAT3*	- Participation in myocardial fibrosis; - Increased expression in patients with liver fibrosis; - Human umbilical cord stem cells.	[54,55]
hsa-miR-29c-3p	100%	*SPARC*	Negative	- Colon cancer.	[56]
hsa-miR-30a-5p	90%	TGF-β-activated kinase 1/MAP3K7-binding protein 3 (TAB3)	Negative on *TAB3*, *α-SMA*, and fibronectin expression	- Bronchoalveolar lavage fluid from idiopathic pulmonary fibrosis patients.	[57]
hsa-miR-34a-5p	90%	*MMP2*	Negative	- Suppressor of breast cancer cells; - Probable tumor suppressor via cell motility regulation of bladder cancer cells.	[58,59]
hsa-miR-92a-3p	100%	*HDAC2*	Not found	- Cartilage tissue; - Chondrocytes.	[60]
hsa-miR-129-5p	70%	Not found	Not found	- Posterior urethral stricture caused by pelvic fracture; - Urethral distraction defects; - Overexpression in scar tissue.	[61]
hsa-miR-132-3p	90%	*ACAN*, *SOX9*, *COL2A1*, *ADAMTS5*	Positive	- Presence in rat mesenchymal stem cells during chondrogenic differentiation.	[62]
hsa-miR-143-3p	80%	*PTGS2*, *MRGPRE*, *PGD2*, *TNF*	Negative	- Rheumatoid arthritis (RA); - Probably useful for the treatment of RA pain; - Negative regulation of pain-associated target genes (*PTGS2*, *TNF*, and *MRGPRE*).	[63]
hsa-miR-155-5p	100%	*CTHRC1* and regulation of Wnt/β-catenin signaling pathway by GSK-3β	Positive	- Hepatocellular carcinoma; - Promotion of kidney fibrosis by SOCS1 and SOCS6 targeting.	[64,65]
hsa-miR-199a-5p	100%	Acting by inhibiting the HIF1a pathway	Not found	- Probable inhibition of the proliferation of keloid fibroblasts; - Promotion of the proliferation of human mesenchymal stem cells; - Scar tissue.	[66,67]
hsa-miR-199b-5p	70%	Cadherin N	Negative on EMT	- Significant inhibition of cell proliferation and invasion of triple-negative breast cancer cells; - Hepatocellular carcinoma.	[68]
hsa-miR-382-5p	100%	*COL1A1*	Positive	- Promotion of epidural fibrosis; - Endometrial cancer.	[26,27]
hsa-miR-661	100%	*MDM2*, *CDC34*	Negative	- Ishikawa endometrial epithelial cells; - Primary human endometrial epithelial cells.	[69]
hsa-miR-664a-5p	80%	Likely impact on MEG3 and SNORA36	Negative	- Osteosarcoma; - Human osteoblast cell line.	[70]
PART C	miRNAs not detected in *OI* nor control cells				
hsa-miR-1-3p	0%	*SOX9*, indirect effect on type X collagen	Negative	- Developmental dysplasia of the hip (DDH); - Chondrocytes.	[71]
hsa-miR-141-3p	0%	*TUG1*, β-catenin	Not found	- Inhibition of epithelial–mesenchymal transition in renal tubular epithelial cells.	[72,73]
hsa-miR-142-3p	0%	*ROCK2*, *CFL2*, *RAC1*, *WASL*	Negative	- Reduction in the migration and contractility of endometrial and endometriotic stromal cells.	[74,75]
hsa-miR-200a-3p	0%	Not found	Not found	- Association with synthesis of ECM components in fibrotic diseases.	[22]
hsa-miR-217	0%	*PTEN*, *DKK1*	Negative on *PTEN* and *DKK1;* positive on *RUNX2* and *COL1A1*	- Promotion of cell proliferation and osteogenic differentiation during development of steroid-associated osteonecrosis in bone marrow mesenchymal stem cells.	[20,21]

**Table 3 pharmaceuticals-16-01414-t003:** Fold change in miRNA expression in *OI* vs. controls.

miRNA Type	Fold Change in Expression in *OI* vs. Control Cells
miR-449a *	−20.9119
miR-19b-3p *	−4.74695
miR-19a-3p *	−4.10178
miR-204-5p *	−3.71922
miR-216a-5p *	−2.52801
miR-10b-5p *	−2.38378
miR-17-5p	−1.91123
miR-382-5p	−1.87937
miR-10a-5p	−1.67714
miR-199a-5p	−1.66411
miR-155-5p	−1.40347
miR-92a-3p	−1.36391
let-7d-5p	−1.34304
let-7a-5p	−1.01942
miR-661	−1.00696
miR-143-3p	1.01308
miR−132-3p	1.23114
miR-34a-5p	1.47401
miR-29a-3p	1.52705
miR-27a-3p	1.54248
miR-26b-5p	1.58722
miR-26a-5p	1.68471
miR-21-5p	1.73538
miR-29c-3p	1.76541
miR-29b-3p	1.78077
miR-25-3p	1.90594
miR-27b-3p	1.93489
miR-30a-5p *	2.20228
miR-199b-5p *	2.89487
miR-129-5p *	3.64773
miR-664a-5p *	6.02516

Negative values show upregulated miRs, and positive values show downregulated miRs in *OI* vs. control cells. * The miRNAs presented on the scatter plot (Figure 1).

**Table 4 pharmaceuticals-16-01414-t004:** Spearman coefficient for miRNAs overexpressed in *OI*. Moderate correlation (0.4–0.6), light-green background; strong correlation (0.6–0.9), green background; complete correlation (1), yellow background; *p*-values are depicted in Appendix A—Appendix A.

	miR-10a-5p	miR-10b-5p	miR-17-5p	miR-199a-5p	miR-19a-3p	miR-19b-3p	miR-204-5p	miR-216a-5p	miR-382-5p	miR-449a
**miR-10a-5p**	1	0.60	0.20	0.50	−0.16	−0.02	0.26	0.09	0.60	−0.14
**miR-10b-5p**	0.60	1.00	0.10	0.67	0.06	−0.05	0.18	−0.06	0.53	−0.01
**miR-17-5p**	0.20	0.1	1.00	0.49	0.08	0.27	0.15	0.09	0.26	0.31
**miR-199a-5p**	0.49	0.67	0.49	1.00	−0.02	−0.20	−0.11	−0.19	0.77	0.11
**miR-19a-3p**	−0.16	0.06	0.08	−0.02	1.00	0.63	0.26	0.34	−0.03	0.74
**miR-19b-3p**	−0.02	−0.05	0.27	−0.20	0.63	1.00	0.66	0.57	−0.27	0.75
**miR-204-5p**	0.26	0.18	0.15	−0.11	0.26	0.66	1.00	0.43	−0.11	0.47
**miR-216a-5p**	0.09	−0.06	0.09	−0.19	0.34	0.57	0.43	1.00	−0.15	0.31
**miR-382-5p**	0.60	0.53	0.26	0.77	−0.03	−0.27	−0.11	−0.15	1.00	0.02
**miR-449a**	−0.14	−0.01	0.31	0.11	0.74	0.75	0.47	0.31	0.02	1.00

**Table 5 pharmaceuticals-16-01414-t005:** Spearman coefficient for miRNAs downregulated in *OI*. Moderate correlation (0.4–0.6), light-green background; strong correlation (0.6–0.9), green background; complete correlation (1), yellow background; moderate negative correlation ((−0.6)–(−0.4)), light-blue background; *p*-values are depicted in Appendix A—Appendix A.

	miR-199b-5p	miR-21-5p	miR-25-3p	miR-26a-5p	miR-26b-5p	miR-27a-3p	miR-27b-3p	miR-29a-3p	miR-29b-3p	miR-29c-3p	miR-30a-5p	miR-664a-5p	miR-129-5p
**miR-199b-5p**	1.00	0.19	0.26	0.31	0.18	0.34	0.32	0.26	0.95	0.27	0.48	0.64	0.49
**miR-21-5p**	0.19	1.00	0.92	0.93	0.92	0.91	0.92	0.95	0.14	0.94	0.69	0.34	−0.42
**miR-25-3p**	0.26	0.92	1.00	0.93	0.92	0.92	0.91	0.92	0.20	0.91	0.79	0.48	−0.31
**miR-26a-5p**	0.31	0.93	0.93	1.00	0.95	0.94	0.94	0.97	0.23	0.96	0.81	0.42	−0.38
**miR-26b-5p**	0.18	0.92	0.92	0.95	1.00	0.91	0.91	0.96	0.09	0.96	0.78	0.32	−0.48
**miR-27a-3p**	0.34	0.91	0.92	0.94	0.91	1.00	0.90	0.92	0.27	0.90	0.86	0.53	−0.27
**miR-27b-3p**	0.32	0.92	0.91	0.94	0.91	0.90	1.00	0.93	0.23	0.92	0.74	0.44	−0.36
**miR-29a-3p**	0.26	0.95	0.92	0.97	0.96	0.92	0.93	1.00	0.16	0.99	0.76	0.35	−0.50
**miR-29b-3p**	0.95	0.14	0.20	0.23	0.09	0.27	0.23	0.16	1.00	0.18	0.41	0.61	0.60
**miR-29c-3p**	0.27	0.94	0.91	0.96	0.96	0.90	0.92	0.99	0.18	1.00	0.73	0.31	−0.49
**miR-30a-5p**	0.48	0.69	0.79	0.81	0.78	0.86	0.74	0.76	0.41	0.73	1.00	0.65	−0.08
**miR-664a-5p**	0.64	0.34	0.48	0.42	0.32	0.53	0.44	0.35	0.61	0.31	0.65	1.00	0.41
**miR-129-5p**	0.49	−0.42	−0.31	−0.38	−0.48	−0.27	−0.36	−0.50	0.60	−0.49	−0.08	0.41	1.00

**Table 6 pharmaceuticals-16-01414-t006:** List of downregulated and overexpressed miRNAs analyzed individually for each *OI* donor. The table depicts the expression trends of miRNAs vs. controls.

Patient Code	Downregulated miRNAs	Upregulated miRNAs
1	miR-10a-5p, miR-10b-5p, miR-19a-3p, miR-26b-5p, miR-34a-5p, miR-199a-5p, miR-382-5p	miR-17-5p, miR-29b-3p, miR-129-5p, miR-143-3p, miR-155-5p, miR-199b-5p, miR-664a
2	let-7a-5p, let-7d-5p, miR-10b-5p, miR-30a-5p, miR-155-5p, miR-199a-5p, miR-382-5p, miR-449a, miR-661	miR-29b-3p, miR-34a-5p, miR-129-5p, miR-132-3p, miR-199b-5p, miR-664a-5p
3	let-7a-5p, miR-10a-5p, miR-10b-5p, miR-21-5p, miR-26b-5p, miR-27a-3p, miR-29a-3p, miR-29c-3p, miR-92a-3p, miR-382-5p	miR-17-5p, miR-29b-3p, miR-129-5p, miR-143-3p, miR-199b-5p, miR-664a-5p, miR-661
4	miR-10a-5p, miR-10b-5p, miR-17-5p, miR-19a-3p, miR-19b-3p, miR-143-3p, miR-199a-5p, miR-449a, miR-661	miR-26a-5p, miR-26b-5p, miR-29a-3p, miR-29c-3p
5	miR-17-5p, miR-19b-3p	let-7a-5p, let-7d-5p, miR-21-5p, miR-25-3p, miR-26a-5p, miR-26b-5p, miR-27a-3p, miR-27b-3p, miR-29a-3p, miR-29c-3p, miR-30a-5p, miR-34a-5p, miR-92a-3p, miR-129-5p, miR-132-3p, miR-155-5p, miR-199a-5p, miR-199b-5p, miR-382-5p, miR-661, miR-664a-5p
6	let-7d-5p, miR-10b-5p, miR-21-5p, miR-26b-5p, miR-27b-3p, miR-29a-3p, miR-34a-5p, miR-92a-3p, miR-132-3p, miR-143-3p, miR-199a-5p, miR-449a	miR-17-5p, miR-29b-3p, miR-30a-5p, miR-129-5p, miR-199b-5p, miR-661, miR-664a-5p
7	miR-10b-5p, miR-19a-3p, miR-19b-3p, miR-199a-5p, miR-449a	miR-10a-5p, miR-21-5p, miR-26a-5p, miR-26b-5p, miR-27a-3p, miR-27b-3p, miR-29a-3p, miR-29c-3p, miR-30a-5p, miR-664a-5p
8	miR-10b-5p, miR-17-5p, miR-19a-3p, miR-19b-3p, miR-129-5p, miR-155-5p, miR-199a-5p, miR-449a, miR-661	let-7a-5p, miR-21-5p, miR-25-3p, miR-26a-5p, miR-26b-5p, miR-27a-3p, miR-27b-3p, miR-29a-3p, miR-29c-3p, miR-30a-5p, miR-34a-5p, miR-664a-5p
9	let-7a-5p, let-7d-5p, miR-10a-5p, miR-10b-5p, miR-19a-3p, miR-19b-3p, miR-132-3p, miR-129-5p, miR-155-5p, miR-199a-5p, miR-382-5p, miR-449a, miR-661	miR-21-5p, miR-26b-5p, miR-27a-3p, miR-27b-3p, miR-29a-3p, miR-29c-3p, miR-30a-5p, miR-664a-5p
10	miR-19a-3p, miR-19b-3p, miR-25-3p, miR-382-5p, miR-449a	miR-27a-3p, miR-29b-3p, miR-30a-5p, miR-34a-5p, miR-129-5p, miR-132-3p, miR-199b-5p, miR-664a-5p

## Data Availability

The data that support the findings of this study are available from the corresponding author upon reasonable request.

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
