# Peer review of "Analysis of miRNAs in Osteogenesis imperfecta Caused by Mutations in COL1A1 and COL1A2: Insights into Molecular Mechanisms and Potential Therapeutic Targets"

_pharmaceuticals, 2023, doi:10.3390/ph16101414_

Round 1

Reviewer 1 Report

1. In Table 2, keep hsa-miR-19b-3p in a single row and state its characters in disease, instead of showing it in two separate rows 2. In Table 2, mention its effect as upregulated or downregulated, rather than mentioning positive or negative 3. Authors need to present their cell viability and qRT-PCR results in the manuscript

Minor corrections

Author Response

Comment 1:  In Table 2, keep hsa-miR-19b-3p in a single row and state its characters in disease, instead of showing it in two separate rows

Response 1: In this case, we meant two separate types of miRs, i.e., hsa-miR-19a-3p (first row) and hsa-miR-19b-3p (second row), this is why it was presented in two separate rows.

Comment 2: In Table 2, mention its effect as upregulated or downregulated, rather than mentioning positive or negative 3. Authors need to present their cell viability and qRT-PCR results in the manuscript

Response 2: Unfortunately, many of the papers referred in the table do not present miRNA the expression data; instead, they focus on either, the impact of a specific miRNA on specific disease or the effect of introducing a specific miRNA into the tested cell lines. To ensure consistency, we have implemented an uniform nomenclature that clearly delineates the influence of miRNAs on genes.

Comment 3: Authors need to present their cell viability and qRT-PCR results in the manuscript.

Response 3: As stated in Section 5.2. Isolation Procedure of Skin Fibroblasts from OI Donors, we achieved a cell viability of approximately 98% using Trypan blue staining. While we have provided graphical representations of the qRT-PCR results in the supplementary file. Additionally, the raw data can be presented by corresponding author upon a reasonable request (e-mail address stated in the manuscript) and/or present on-line in the repository site.

Reviewer #2

Comment:

This is a potentially interesting paper that approaches changes in miRNA expression in osteogenesis imperfecta. The authors reported increased expression of six miRNAs, and decreased expression of four miRNAs.

One of the first papers to investigate the expression of miRNAs in osteogenesis imperfecta was published in 2014, by Kaneto et al (BMC Medical Genetics 2014, 15:45), which reported decreased expression of miR-29b. The paper was cited (26), but the different results obtained here were not discussed. A brief discussion should be included, maybe around lines 299-303, comparing the results here presented to others, already published.

The presentation of results in Tables is not easy. Maybe the more relevant data should be also presented in graphics, to make easier the visualization of the most important results.

Reviewer 2 Report

This is a potentially interesting paper that approaches changes in miRNA expression in osteogenesis imperfecta. The authors reported increased expression of six miRNAs, and decreased expression of four miRNAs.

One of the first papers to investigate the expression of miRNAs in osteogenesis imperfecta was published in 2014, by Kaneto et al (BMC Medical Genetics 2014, 15:45), which reported decreased expression of miR-29b. The paper was cited (26), but the different results obtained here were not discussed. A brief discussion should be included, maybe around lines 299-303, comparing the results here presented to others, already published.

The presentation of results in Tables is not easy. Maybe the more relevant data should be also presented in graphics, to make easier the visualization of the most important results.

Author Response

Response:

Regarding the findings from Kaneto et al.'s study, we sincerely appreciate your observation regarding the absence of a comparison with our results. We have included a brief discussion addressing this issue (lines 299-305).

While considering different options for presenting the results, we encountered the challenge of managing a large volume of data. As a result, we opted for the most transparent approach for us. We have placed the results in graphical form in the supplementary file (Supplementary Fig. 1). If you have any suggestions or insights on how we can further enhance the visualization of our results, we would genuinely welcome your guidance and assistance.

Reviewer 3 Report

The authors analyzed expression of 40 miRNAs in skin fibroblasts of OI 10 patients with OI as a causative agents of the disease and identified 9 miRNAs related only to OI patients.

Comments

1.      Lines 37-40: The authors Conclusion is not clear. It should be clarified.

2.      Lines 127-132 should be moved to Introduction section.

3.      Lines 146-151; 178-181 should be moved to the Discussion section.

4.      Table 2, part B: The authors should indicate in separate sections the miRNAs, which were upregulated or downregulated in patients with OI compared to controls.

5.      The same data is presented in Fig 1 and Table 3. The authors should not duplicate the same results. There is not reference to Table 3 in the text. This should be corrected.

6.      Table 4 and 5 should be supplemented with p=values to each correlation coefficient.

7.      Lines 265, 280, 285, 297,299: references are required at the end of these serntences.

8.      Lines 313-315: This sentence repeats that in the previous paragraph. This should be corrected.

9.      Conclusion is missing. This should be corrected.

10.  Lines 295-298: The authors should present a rationale for this statement as it is not clear.

11.  The references should be organized according to the Journal requirements.

1.      Lines 37-40: The authors Conclusion is not clear. It should be clarified.

1.      Lines 295-298: The authors should present a rationale for this statement as it is not clear.

Author Response

Comment 1: Lines 37-40: The authors Conclusion is not clear. It should be clarified.

Response 1: The conclusion in 4.6.Conclusion section was corrected according to the Reviewers suggestion. The conclusion in the abstract was not changed due to the word limit.

Comment 2:  Lines 127-132 should be moved to Introduction section

Response 2: Lines 127-132 have been moved to the Introduction section on line 77 as suggested.

Comment 3: Lines 146-151; 178-181 should be moved to the Discussion section

Response 3: Lines 146-151 and 178-181 have been moved to the Discussion section on line 260 and 288 (respectively) according to the Reviewers suggestion.

Comment 4: Table 2, part B: The authors should indicate in separate sections the miRNAs, which were upregulated or downregulated in patients with OI compared to controls

Response 4: Our intention was to prevent redundancy of presented results. In Table 3 we categorized miRNAs as either, upregulated (negative values) or downregulated (positive values), whereas Table 6 separates miRNAs based on individual patient data. To enhance the overall readability, the content of Table 3 was revised.

Comment 5: The same data is presented in Fig 1 and Table 3. The authors should not duplicate the same results. There is not reference to Table 3 in the text. This should be corrected

Response 5: We added link to the Table 3 as suggested. It's worth noting that there is some overlap in the results, which we included deliberately to highlight the confirmation of expression changes. Specifically, Table 3 encompasses the entire list of tested miRNAs, while the scatter plot displays exclusively those with notable and statistically significant alterations in expression.

Comment 6: Table 4 and 5 should be supplemented with p=values to each correlation coefficient

Response 6: Certainly, we have included the p-values, but to ensure the transparency of our results, it was presented in the supplementary file (Supplementary Table 2).

Comment 7: Lines 265, 280, 285, 297,299: references are required at the end of these serntences

Response 7: We added references to the lines 265, 280, 285, 297 as suggested. Line 299 has a reference at the end of the sentence. Additionally, the typo miR-644a-5p to miR-664a-5p on line 278 was corrected.

Comment 8: Lines 313-315: This sentence repeats that in the previous paragraph. This should be corrected

Response 8: We were unable to localize the repeated sentence in previous paragraph.

Comment 9: Conclusion is missing. This should be corrected

Response 9: Conclusion was corrected according to the Reviewers suggestion.

Comment 10: Lines 295-298: The authors should present a rationale for this statement as it is not clear

Response 10: The rationale (lines 295-298) has been included in lines 298-305.

Comment 11: The references should be organized according to the Journal requirements

Response 11: The literature was organized in accordance with the journal's requirements.